# Gene editing therapy as a therapeutic approach for cardiovascular diseases in animal models: A scoping review

Quan Duy Vo[1,2]*

1 Department of Cardiovascular Medicine, Faculty of Medicine, Dentistry and Pharmaceutical Sciences, Okayama University, Okayama Japan, 2 Faculty of Medicine, Nguyen Tat Thanh University, Ho Chi Minh, Viet Nam

* dr.duyquan@gmail.com

## Abstract

### Background

Cardiovascular diseases (CVDs) are the leading cause of mortality worldwide, with hereditary genetic factors contributing substantially to disease burden. Current treatments, including lifestyle modifications, pharmacotherapy, and surgical interventions, focus primarily on symptom management but fail to address underlying genetic causes, often resulting in disease progression or recurrence. Gene therapy has emerged as a transformative approach, offering a potential treatment. This review explores its efficacy and safety in animal models, identifying opportunities for future advancements.

### Methods

This review investigated studies on gene editing interventions in animal models of CVDs, retrieved from PubMed, ScienceDirect, and Web of Science up to December 2024.

### Result

A total of 57 studies were included in this review. Mice (86%) were the predominant model, with CRISPR-Cas9 (53%) and AAV vectors (80%) as the most used tools. Key targets included *PCSK9* (32%), *LDLR* (9%), and *MYH6/7* (7%), achieving 25–85% editing efficiency in liver/heart tissues. Base editors (ABE/CBE) showed superior safety, with <1% off-targets versus CRISPR-Cas9's 2–5 off-targets per guide. Reported toxicity risks included liver injury (AAVs, 23%) and transient cytokine elevation (LNPs, 14%).

**Data availability statement:** All relevant data are within the manuscript and its Supporting Information files.

**Funding:** The author(s) received no specific funding for this work.

**Competing interests:** The authors have declared that no competing interests exist

**Abbreviations:** AAV, adeno-associated viruses; ABE, Adenine base editor; AdV, Adenoviral vector; ALT, Alanin aminotransferase; ANGPTL3, Angiopoietin-like protein 3; ARRIVE, Animal Research: Reporting of In Vivo Experiments; ASGR, asialoglycoprotein receptor; AST, Aspartat aminotransferase; BE, Base editors; CRISPR-Cas9, Clustered Regularly Interspaced Short Palindromic Repeats-associated protein 9; CVD, Cardiovascular diseases; FH, Familial hypercholesterolemia; gRNA, guide RNA; HDR, homology-directed repair; LDL-c, Low density lipoprotein cholesterol; LDLR, Low density lipoprotein receptor; MeSH, Medical Subject Headings; NHEJ, non-homologous end joining; PAM, Protospacer adjacent motif; PRISMA, Preferred Reporting Items for Systematic Reviews and Meta-Analyses; PSCK9, Proprotein convertase subtilisin/kexin type 9; TALEN, Transcription activator-like effector nuclease; TTR, Transthyretin; ZFN, Zinc-finger nuclease.

## Conclusion

Gene editing therapy shows great potential for treating CVDs, with high efficiency, strong therapeutic outcomes, and favorable safety in animal models. Continued innovation and rigorous evaluation could transform cardiovascular treatment, benefiting patients with untreatable conditions.

## Introduction

Cardiovascular diseases (CVDs) are the leading cause of mortality worldwide, accounting for approximately 17.9 million deaths annually, which represents 32% of all global deaths [1]. In the United States alone, heart disease is responsible for about 697,000 deaths each year, equating to one in every five deaths. The prevalence of CVDs increases with age, affecting 24.2% of adults aged 75 and over [2]. Despite remarkable progress in preventive measures and treatments, CVD prevalence continues to rise due to population aging and the increasing prevalence of associated risk factors such as hyperlipidemia, hypertension, and diabetes [3]. Importantly, a substantial proportion of CVD cases have a hereditary component, driven by genetic factors that predispose individuals to conditions like familial hypercholesterolemia, hypertrophic cardiomyopathy, and inherited arrhythmias [4,5]. Traditional therapeutic approaches for CVDs primarily focus on managing symptoms and mitigating risk factors. These include lifestyle modifications, pharmacotherapy to control blood pressure and cholesterol levels, and surgical interventions to address structural heart issues. While these strategies have proven beneficial, they often require lifelong adherence and may not address the underlying genetic causes of the disease [6].

Gene therapy has emerged as a transformative approach for addressing both inherited and acquired cardiovascular conditions [7]. By enabling precise modifications to the genetic basis of these diseases, gene therapy provides the potential to correct or mitigate genetic mutations responsible for cardiovascular disorders such as hypertrophic cardiomyopathy and familial hypercholesterolemia. Early therapies introduced functional gene copies to restore normal function. However, these efforts faced significant challenges, including limited delivery efficiency and immune responses that hindered their effectiveness [8]. The advances in vector technology, particularly the development of adeno-associated viruses (AAVs), significantly improved the safety, specificity, and efficiency of gene delivery to cardiovascular tissues [9]. A pivotal advancement in the gene therapy landscape is the introduction of CRISPR-Cas9 (Clustered Regularly Interspaced Short Palindromic Repeats-associated protein 9). These technologies enable precise editing of specific genomic sequences, offering a groundbreaking approach to directly correct disease-causing mutations at their source and thereby address the underlying genetic causes of cardiovascular diseases [10].

Fundamentally, gene editing distinguishes itself from conventional therapies through its target specificity and durability. Unlike traditional treatments that continuously address symptoms without correcting the genetic causes, gene editing offers

a permanent, one-time correction of pathogenic genetic variants [11]. Given these considerable advancements, there is growing interest in translating gene editing technologies into viable therapeutic strategies for cardiovascular diseases. This review aims to evaluate the current landscape of gene editing applications in the treatment of cardiovascular diseases using animal models. By synthesizing existing research, we seek to elucidate the therapeutic potential, efficacy, and safety of gene editing interventions in preclinical settings. Furthermore, this review will identify existing knowledge gaps and propose directions for future research, ultimately contributing to the advancement of gene-based therapies for cardiovascular diseases.

## Materials and methods

This review was conducted in accordance with the Preferred Reporting Items for Systematic Reviews and Meta-Analyses (PRISMA) guidelines, ensuring a transparent and comprehensive synthesis of existing literature [12].

### Eligibility criteria

The inclusion criteria encompassed original research articles that examined gene editing interventions, including CRISPR-Cas9, transcription activator-like effector nucleases (TALENs), and zinc finger nucleases (ZFNs), in animal models of CVDs. Eligible studies were required to report on therapeutic efficacy, safety, and outcomes. Articles were excluded if they were not original research (review articles, conference abstracts/posters), did not involve gene editing for therapeutic purposes, or were published in languages other than English.

### Search strategy

A systematic literature search was conducted across multiple databases, including PubMed, Embase, and Web of Science, covering publications up to December 2024. The search strategy incorporated a combination of keywords and Medical Subject Headings (MeSH) terms related to gene editing (e.g., CRISPR, base editing, prime editing, zinc finger nucleases, and TALEN), cardiovascular diseases (e.g., heart disease and cardiomyopathy), and animal models (e.g., mouse, rat, pig, and non-human primates).

In addition to database searches, relevant studies were identified through manual reference screening, ensuring inclusion of key publications that may not have been captured by the initial search strategy.

### Data extraction

Data extraction was performed using a standardized form to collect key information from each study, including authorship, publication year, animal model species, sample size, gene editing technology, target genes, delivery methods, therapeutic efficacy, safety outcomes, and main findings.

### Quality assessment

The methodological quality of the included studies was evaluated using the Animal Research: Reporting of In Vivo Experiments (ARRIVE) guidelines, focusing on aspects such as study design, statistical analysis, and ethical considerations [13].

## Result

Initially, a total of 995 articles were identified from three electronic databases. After removing 152 duplicates, 843 studies proceeded to title and abstract screening based on the predefined inclusion criteria. Following this screening process, 201 articles were deemed eligible for full-text evaluation. However, a thorough review led to the exclusion of 144 studies, ultimately including 57 studies in this review (Fig 1).

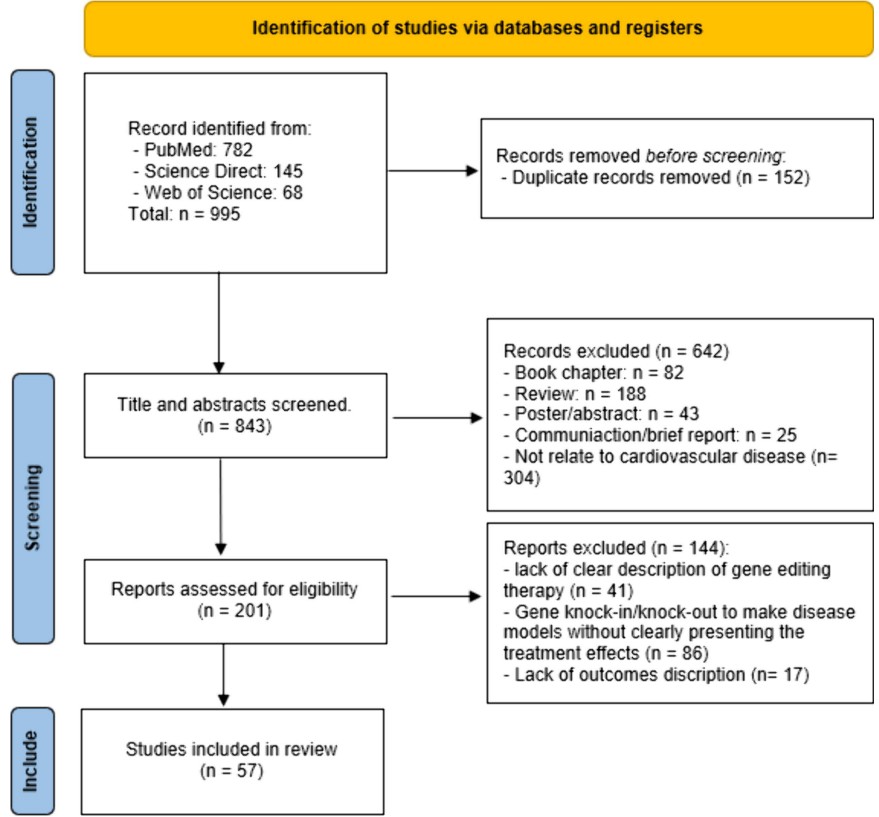

**Fig 1. Study flow-chart.**

## Overview of included studies

A total of 57 studies were included in this review, evaluating the role of gene editing in cardiovascular disease models. The majority of studies (86%, 43/50) used mice, reflecting their cost-effectiveness, genetic tractability, and well-characterized cardiovascular physiology. Larger animal models, such as non-human primates (NHPs, 7%) and pigs (5.2%), were primarily employed for translational validation (Table 1).

CRISPR-Cas9 was the most prevalent gene editing method, used in 30 (53%) of studies. Its dominance is largely attributed to its high efficiency and broad versatility in genome editing applications. Base editors, including adenine base editors (ABEmax, ABE8e) and cytidine base editors (CBE such as BE3), accounted for 20% of the studies. These tools have gained increasing attention due to their ability to introduce precise single-nucleotide changes without generating double-strand breaks. Other systems were used less frequently, including meganucleases (2%) and RNA-targeting CRISPR variants like CRISPR-Cas13d (2%), typically applied in specialized contexts requiring high specificity or RNA-level editing. A small number of studies also investigated alternative approaches such as CRISPR-Cas10 or gene silencing strategies like shRNA knockdown.

Among gene delivery methods, adeno-associated virus (AAV) vectors emerged as the predominant delivery system for in vivo gene editing, utilized in 80% of the included studies. Among these, AAV9 (40%) and AAV8 (25%) were the most frequently employed serotypes, primarily due to their strong tissue tropism for the liver and heart. Non-viral delivery approaches, particularly lipid nanoparticles (LNPs), were implemented in approximately 19% of studies, reflecting growing interest in transient and efficient gene transfer methods that avoid the risks associated with viral integration. Adenoviral

**Table 1. Distribution of animal models.**

| Species/Strain | Frequency | Percentage (%) | Notes |
|---|---|---|---|
| **Mice** | 49 | 86% | Most common: C57BL/6, LDLR$^{-/-}$, ApoE$^{-/-}$ |
| **Rats** | 2 | 3.5% | Sprague-Dawley (1), MYL4-E11K (1) |
| **Hamsters** | 2 | 3.5% | LDLR$^{-/-}$ model |
| **Rabbits** | 2 | 3.5% | LDLR$^{-/-}$ (1), APOC3-KO (1) |
| **Pigs** | 3 | 5.2% | Domestic pigs, Yucatan mini-pigs |
| **Non-Human Primates (NHPs)** | 4 | 7% | Cynomolgus macaques (3), Rhesus (1) |

vectors were used in 9% of studies, although their application has diminished in recent years owing to immunogenicity concerns. Other delivery methods, including plasmid electroporation, microinjection (commonly for embryonic gene editing), and virus-like particles, were employed only in a limited number of cases. (Fig 2).

## Gene selection and target organs

The most frequently edited gene was PCSK9 (32%, 18 studies), a key regulator of LDL cholesterol, reflecting its prominence in hypercholesterolemia models. Beyond metabolic targets, several studies focused on genes implicated in cardiomyopathies and heart failure. *MYH6* and *MYH7*, which encode cardiac myosin heavy chains and are frequently mutated in hypertrophic cardiomyopathy (HCM), were edited in 10% of studies to correct sarcomeric dysfunction. Other notable targets include *PLN* (phospholamban), involved in calcium handling abnormalities in dilated cardiomyopathy (DCM), and *RBM20*, a gene associated with RNA splicing defects in heart failure. In addition, muscle-related disorders were also addressed, with *DMD* (dystrophin) targeted in 6% of studies aiming to restore protein expression in Duchenne muscular dystrophy-associated cardiomyopathy. The distribution of these gene-editing efforts reflected organ-specific priorities, with the liver and heart being the primary targets, consistent with the physiological relevance of the genes involved.

## Gene editing efficiency

Gene editing efficiency – defined as the proportion of cells or alleles successfully edited – varied across studies depending on the editing tool and delivery method. Table 1 summarizes these efficiencies alongside each study's target gene, editing platform, and delivery strategy, with values representing the maximum in vivo editing achieved.

CRISPR-Cas9 generally produced high editing efficiencies, particularly in liver targets, with indel rates commonly ranging from 40% to 85%, and in some cases reaching up to ~95%. For instance, liver editing efficiencies reached 67% in *PCSK9* [14], while cardiac editing, such as *MYH6* inactivation, achieved up to 72% [15]. Base editors (ABE/CBE) showed moderate but therapeutically relevant DNA editing rates (typically 15–70%), with some studies reporting up to 63% efficiency [16]. RNA editing efficiencies were even higher, reaching up to 99,2% in SCN5a correction [17].

Delivery methods also played a critical role. AAVs yielded high and stable editing in liver and heart tissues, though efficiency was dose-dependent, requiring doses between $1 \times 10^{11}$ and $1 \times 10^{13}$ vg/kg. LNPs, while offering rapid hepatocyte uptake and detectable editing within 48 hours, typically provided transient expression (Table 2).

## Therapeutic outcomes

Despite varying editing efficiencies, all studies reported beneficial therapeutic outcomes in their respective disease models. In hyperlipidemia studies, targeting genes such as *PCSK9*, *LDLR*, and *ANGPTL3* consistently resulted in significant reductions in serum cholesterol levels [14,18,19,27]. In cardiomyopathy models (both hypertrophic and dilated), gene

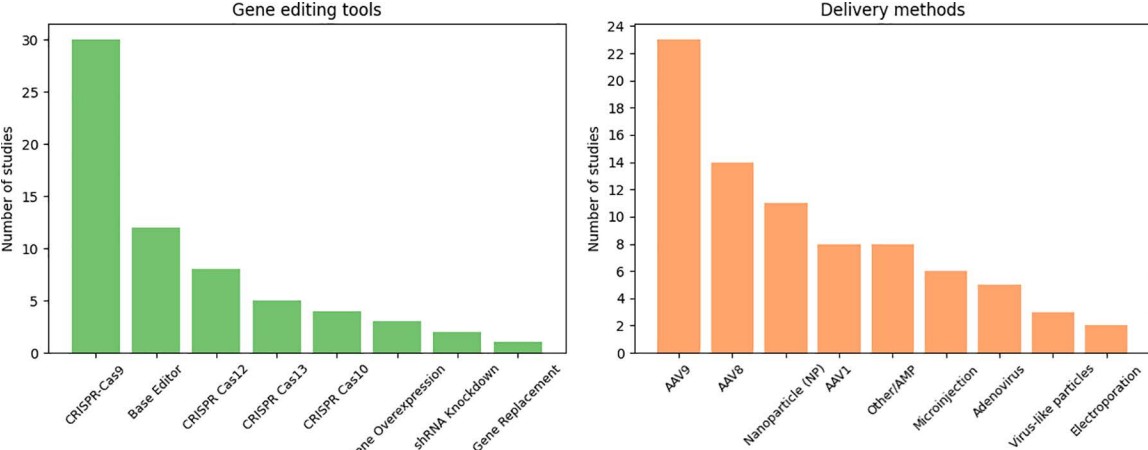

**Fig 2. Gene editing system employed in the included studies.**

editing led to improved cardiac function, such as increased ejection fraction or reduced myocardial fibrosis, translating to better heart performance [20,21,26]. In the Duchenne muscular dystrophy models, reintroducing dystrophin via gene editing improved muscle strength and function [22,37,67]. Arrhythmia models (Long QT syndrome [17], atrial fibrillation [24], catecholaminergic polymorphic VT [64]) presented correction of electrical abnormalities, evidenced by normalized ECG parameters or fewer arrhythmic episodes.

Metabolic disease models like hereditary tyrosinemia type I showed restoration of metabolic function and survival [25]. Likewise, the transthyretin amyloidosis model demonstrated reduced pathogenic *TTR* protein deposition [32]. Importantly, many of these therapeutic benefits were the direct result of the gene editing event – for example, base editing of *PCSK9* in primates led to durable cholesterol lowering, and editing *ANGPTL3* in mice halved triglyceride levels, highlighting the potential clinical impact [14,31].

Notably, even modest editing levels were effective in some disease models; for example, ~5–6% dystrophin restoration in Duchenne muscular dystrophy was sufficient to produce functional benefit [22]. These findings underscore that therapeutic impact is not always directly proportional to editing efficiency, especially in models with low correction thresholds.

Recent findings by Feng et al. (2024) identified *MST1R* as a novel gene associated with Tetralogy of Fallot, demonstrating that its loss impairs cardiomyocyte differentiation and contractile function. These results highlight *MST1R* as a promising therapeutic target for gene editing-based treatment of congenital heart disease [71] (Table 3).

## Safety and off-target effects

Gene editing therapies showed a favorable safety profile in animal models. Many studies used genome-wide analyses to assess off-target effects, with at least 13 reporting no detectable off-target edits, particularly when high-fidelity or base editors were used. Base editing demonstrated superior precision – for example, no off-target mutations or chromosomal translocations were observed in a humanized PCSK9 model, in contrast to low-frequency off-target events and structural variants with standard CRISPR-Cas9 [18].

Among studies reporting off-target analyses (30 studies), detected edits were typically rare and located in non-coding regions, with no cases severe enough to negate therapeutic benefits. GUIDE-seq identified 2–5 off-target sites per guide in Cas9 systems [18], and AAV integration at cut sites was noted in up to 30% of cases [48]. Base editors had < 1% off-target activity, with occasional bystander edits reported [62].

**Table 2. Gene editing efficiency.**

| Study | Target Gene | Gene Editing Tool | Delivery Method | Editing Efficiency |
|---|---|---|---|---|
| Alba Carreras, 2019 [18] | PCSK9 | Cas9, BE3 | AdV | 10–35% (Cas9/BE3) |
| Alexandra C. Chadwick, 2018 [19] | ANGPTL3 | BE3 | AdV | 35% (liver, day 7) |
| Andreas C. Chai, 2023 [20] | MYH7 | ABEmax-VRQR | Dual AAV9 | 35% (DNA), 12.9–26.7% (RNA) |
| Bin Li, 2021 [21] | XIRP1 | Overexpression | AAV9 | N/A (functional rescue only) |
| Camilo Breton, 2020 [14] | PCSK9 | Meganuclease | AAV8 | 67% (mice), 15–43% (NHPs) |
| Chengzu Long, 2016 [22] | DMD | SpCas9 | AAV9 | 25.5% (TA muscle), 70% (heart) |
| Daniel Reichart, 2023 [15] | MYH6 | ABE8e | Dual AAV9 | 81% (cDNA), 16% (gDNA) |
| F Ann Ran, 2015 [23] | PCSK9 | SaCas9 | AAV8 | >40% indels (liver) |
| Handan Hu, 2022 [24] | MYL4 | Overexpression | AAV9 | N/A (protein restored) |
| Hao Yin, 2017 [25] | PCSK9 | e-sgRNA+Cas9 mRNA | LNP | 83% (Pcsk9), >40% (Fah) |
| Hengzhi Du, 2024 [26] | CRT | CRISPR-Cas9 | AAV9 | 63.3% (DNA), 88% (RNA) |
| Huan Zhao, 2020 [27] | LDLR | CRISPR-Cas9 | AAV8 | 6.7% HDR, 25% indels |
| Jaydev Dave, 2022 [28] | PLN | SaCas9 | AAV9 | 72% (LV inactivation) |
| Jessie R. Davis, 2022 [29] | PCSK9/ANGPTL3 | ABE8e | AAV8/AAV9 | 44–61% (liver) |
| Jiacheng Li, 2022 [30] | Meis1/Hoxb13 | CasRx | AAV9 | 65.2% (Meis1), 83.6% (Hoxb13) |
| Jing Gong, 2020 [31] | PCSK9/ANGPTL3 | CRISPR-Cas9 | LipoMSN | 24.8% (Pcsk9), 7.2% (Angptl3) |
| Jonathan D. Finn, 2018 [32] | TTR | SpyCas9 | LNP | ~70% (liver) |
| Jonathan M. Levy, 2020 [33] | NPC1 | ABEmax/CBE3.9max | Dual AAV | 38% (liver, ABE), 21% (CBE) |
| Kelsey E. Jarrett, 2017 [34] | LDLR/ApoB | SpyCas9 | AAV8 | 54.3% (Ldlr), 74.1% (Apob) |
| Kelsey E. Jarrett, 2019 [35] | LDLR | SaCas9 | AAV8 | 31.9% (males), 33.1% (females) |
| Kiran Musunuru, 2021 [16] | PCSK9 | ABE8.8 | LNP | 63–66% (hepatocytes) |
| Lei Huang, 2017 [36] | ApoE/LDLR | CRISPR-Cas9 | Electroporation | N/A (knockout confirmed) |
| Li Xu, 2019 [37] | DMD | Cas10 | AAVrh.74 | 11.1% (cardiomyocytes) |
| Lili Wang, 2021 [38] | PCSK9 | Meganuclease | AAV8/AAV3B | 9.5–64.4% (liver) |
| Lingmin Zhang, 2019 [39] | PCSK9 | CRISPR/Cas9 | Gal-LGCP | 60% (liver) |
| Lisa N. Kasiewic, 2023 [40] | LDLR | ABE8.8 | GalNAc-LNP | 61% (liver) |
| Luzi Yang, 2024 [41] | CaMKIIδ | Adenine Base Editor | AAV9 | >90% reduction transgene-positive cells (liver) |
| Luzi Yang, 2024 [42] | LMNA | Adenine Base Editor | Dual AAV system | 20% at bystander site ~8% at the disease-causing site |
| Man Qi, 2024 [17] | SCN5a | ABEmax | Dual AAV9 | 43.04% (DNA), 99.2% (RNA) |
| Marco De Giorgi, 2021 [43] | ApoA1 | SaCas9 | Dual AAV8 | 54% (indels), 7.8% (HDR) |
| Markus Grosch, 2022 [44] | RBM20 | ABE | AAVMYO | 21.4% (DNA), 71% (RNA) |
| Mengmeng Guo 2020 [45] | ApoC3 | CRISPR/Cas9 | zygote microinjection | Not report |
| Min Qiu, 2021 [46] | ANGPTL3 | SpCas9 | LNP | 38.5% (liver) |
| Ping Yang, 2024 [47] | MYH6 | Cas13d | AAV9 | 27.1–32% (RNA knockdown) |
| Qian Li, 2021 [48] | PCSK9 | SaCas9 | AAV8 | 25–45% (liver) |
| Qiang Cheng, 2020 [49] | PCSK9/PTEN | Cas9 mRNA/RNP | LNP | ~60% (Pcsk9), 14% (PTEN) |
| Qiurong Ding, 2014 [50] | PCSK9 | SpCas9 | AdV | 50% (liver) |
| Richard G. Lee, 2023 [51] | PCSK9 | ABE8.8m | LNP | 46–70% (liver) |
| Rui Lu, 2018 [52] | LDLR | CRISPR-Cas9 | Microinjection | N/A (knockout confirmed) |
| Samagya Banskota, 2022 [53] | PCSK9 | ABEmax | VLPs | 63% (hepatocytes) |
| Shijie Liu, 2021 [54] | Sav | shRNA | AAV9 | N/A (functional rescue) |
| Shuhong Ma, 2021 [55] | MYH6 | ABEmax-NG | Microinjection/AAV9 | 62.5% (embryos), 25.3% (AAV9) |

*(Continued)*

**Table 2.** (Continued)

| Study | Target Gene | Gene Editing Tool | Delivery Method | Editing Efficiency |
|---|---|---|---|---|
| Shuo Wu, 2024 [56] | MYBPC3 | Base editing | Dual AAV9 | 5-10% mutation correction (heart) 38%–100% protein recovery |
| Simon Lebek, 2023 [57] | CaMKIIδ | ABE8e-SpRY | Dual AAV9 | 7.5–8.4% (DNA), 46–85.7% (RNA) |
| Simon Lebek, 2023 [58] | CaMKIIδ | ABE8e-SpCas9-NG | Microinjection | N/A (phenotypic rescue) |
| Simon Lebek, 2024 [59] | CAMK2D | ABE8e | MyoAAV2A | 36.2–37% (DNA), 83.2% (RNA) |
| Suya Wang, 2020 [60] | TAZ | Gene replacement | AAV9 | N/A (protein restored) |
| Takahiko Nishiyama, 2022 [61] | RBM20 | ABEmax-VRQR-SpCas9 | AAV9 | 19% (DNA), 66% (RNA) |
| Tanja Rothgangl, 2021 [62] | PCSK9 | ABEmax | LNP | 58% (mice), 35–40% (NHPs) |
| Xiao Wang, 2017 [63] | PCSK9 | CRISPR-Cas9 | Adenovirus | 40-70 indels (liver) |
| Xiaolu Pan, 2018 [64] | RYR2 | SaCas9 | AAV9 | 11.3% (DNA), 21.1% (RNA) |
| Xin Guo, 2017 [65] | LDLR | CRISPR-Cas9 | Microinjection | N/A (knockout confirmed) |
| Yiwen Zha, 2021 [66] | APOC3 | CRISPR-Cas9 | Microinjection | N/A (knockout confirmed) |
| Yu Zhang, 2020 [67] | DMD | CRISPR-Cas9 | Dual AAV | 50–100% (tissue-dependent) |
| Yuanbojiao Zuo, 2023 [68] | ANGPTL3 | CRISPR-Cas9 | Dual AAV9 | 63.3% (DNA), 88% (RNA) |
| Zhanzhao Liu, 2025 [69] | CAMK2d | Adenine Base Editors | AAV9 | Not mention |
| Zhiquan Liu, 2021 [70] | PCSK9/TYR/MSTN | SpaCas9 | AAV8 | 16.6% (Pcsk9), 40–78% (zygotes) |

**Table 3. Gene targets and therapeutic outcomes.**

| Gene | Associated Disease | Editing Strategy | Efficiency Range |
|---|---|---|---|
| ANGPTL3 | Hypertriglyceridemia | CRISPR-KO, ABE | 40–75% (DNA/RNA) |
| APOC3 | Hyperlipidemia | CRISPR-KO | 50% (Protein) |
| CAMK2D | Heart Failure | ABE | 7.5–85.7% (RNA/DNA) |
| DMD | Duchenne Muscular Dystrophy | Exon Skipping, Frame Restoration | 5–50% (Protein) |
| LDLR | Atherosclerosis, Hyperlipidemia | CRISPR-HDR, NHEJ, Base Editing | 30–70% (DNA) |
| MYH6/7 | Hypertrophic Cardiomyopathy (HCM) | ABE, CRISPR Correction | 20–80% (RNA/DNA) |
| NPC1 | Niemann-Pick Type C | ABE, CBE | 21–38% (DNA) |
| PCSK9 | Hypercholesterolemia | CRISPR-KO, ABE, Meganuclease | 25–85% (DNA/RNA) |
| PLN | Cardiomyopathy/Arrhythmia | CRISPR-KO | 72% (DNA) |
| RBM20 | Dilated Cardiomyopathy (DCM) | ABE | 19–71% (RNA/DNA) |
| RYR2 | CPVT (Arrhythmia) | CRISPR-KO | 11–21% (DNA/RNA) |
| SCN5A | Long QT Syndrome | ABE | 43–99% (RNA/DNA) |
| TAZ | Barth Syndrome | Gene Replacement | ~70% (Protein) |
| TTN | Dilated Cardiomyopathy | Splicing Rescue (via RBM20 editing) | 50% (RNA) |
| TTR | Amyloidosis | CRISPR-KO | ~70% (DNA) |

Toxicity assessments revealed no significant adverse effects. Histological analyses of major organs showed no abnormal inflammation or damage, and liver enzyme levels and immune markers remained within normal ranges. No long-term malignancies or severe outcomes were reported (Table 4).

### Randomization and blinding

Attention to bias reduction in study design was variable. Randomization of animals into treatment versus control groups was reported in 16 out of 57 studies (28%). Blinding of investigators to group allocation during outcome assessment was

**Table 4. Toxicity profiles.**

| Gene Editing Tool | Delivery Method | Key Toxicity Risks | Incidence | Notable Genes |
|---|---|---|---|---|
| Base Editors (CBE/ABE) | AAV vectors or lipid nanoparticles (mRNA+gRNA) | Minimal off-target edits (DNA or RNA base changes); occasional transient hepatotoxicity (mild ALT/AST elevations); mild immune activation (e.g., cytokines) | ~30% studies | *PCSK9, ANGPTL3, LDLR, LMNA, MYBPC3, MYH6, MYH7, RBM20, SCN5A, DNMT1, NPC1, CAMK2D* |
| CRISPR-Cas9 (DNA nuclease) | AAV vectors (most common); also Cas9 RNP or mRNA (e.g., zygote injection) | Off-target indel mutations; on-target large deletions/rearrangements; liver injury (elevated ALT/AST); immune responses (transient cytokine release); AAV vector genome integration at cut sites | ~40% studies | *PCSK9, LDLR, ANGPTL3, APOB, APOA1, APOE, APOC3, CRT, DMD, PLN, PTEN, RYR2, TTR, TYR* |
| CRISPR-Cas10 | AAV vectors | Mild immune response to vector (e.g., low anti-AAV immunity); no significant off-target mutations or organ damage observed | 1 study | *DMD* |
| CRISPR-Cas13 (RNA targeting) | Viral vectors (e.g., AAV) or plasmid delivery | Collateral RNA cleavage (off-target transcript degradation) – generally minimal with optimized Cas13; no overt organ toxicity reported in vivo | 50% studies | *MYH6, MEIS1, HOXB13, MHRT* |
| Meganucleases (e.g., I-CreI variants) | AAV vectors (e.g., AAV8) | Off-target cleavage at unintended sites; liver enzyme elevations (transient liver injury); immune responses (anti-capsid or anti-nuclease T-cell response) | 2 studies | *PCSK9* |

reported in 20 studies (35%). Notably, 10 studies (17.5%) explicitly stated that they implemented both randomization and blinding in their experimental design. Many studies neither mentioned nor clarified these measures, indicating a need for improved methodological rigor in preclinical gene therapy research.

More detailed information about the included studies is provided in Table S1, Supplemental Materials.

## Discussion

### The development of gene editing systems

Gene editing technologies have evolved significantly over recent decades, progressing from early tools such as zinc-finger nucleases (ZFNs) and transcription activator-like effector nucleases (TALENs), which induce double-strand breaks (DSBs) at specific genomic loci, to more advanced and precise systems (Fig 3).

Zinc finger nucleases (ZFNs) employ engineered zinc finger domains that recognize DNA triplets and are coupled to FokI nucleases [72]. TALENs, on the other hand, utilize transcription activator-like effector proteins, each recognizing a single nucleotide, also linked to FokI nucleases [73,74]. This one-to-one recognition allows for more straightforward and precise targeting compared to ZFNs [75,76]. However, these methods were complex to design and required extensive protein engineering, limiting their widespread application [77].

The introduction of the CRISPR-Cas9 system has revolutionized gene editing by providing a more accessible, efficient, and versatile approach. CRISPR-Cas9, originally discovered as an adaptive immune mechanism in bacteria, was repurposed in 2012–2013 as a programmable gene editing system [78]. This platform employs a single-guide RNA (sgRNA) to direct the Cas9 nuclease precisely to complementary DNA sequences, thereby generating site-specific DSBs, followed by repair primarily via non-homologous end joining (NHEJ) or homology-directed repair (HDR) [79]. NHEJ often introduces disruptive insertions or deletions, useful for gene knockout studies like PCSK9 deletion for cholesterol reduction [18]. HDR enables precise mutation correction when a repair template is available, though it remains inefficient in vivo and is limited to dividing cells [80]. A critical consideration in CRISPR-Cas9 design is the requirement for a protospacer adjacent motif (PAM) near the target sequence, which can also contribute to off-target activity [81].

To overcome the limitations of CRISPR-Cas9, derivative technologies like base editing were developed. Base editors combine catalytically impaired Cas enzymes with nucleotide-modifying enzymes to enable precise single-base conversions without inducing double-strand breaks [82]. The main types – cytosine base editors (CBEs) and adenine base

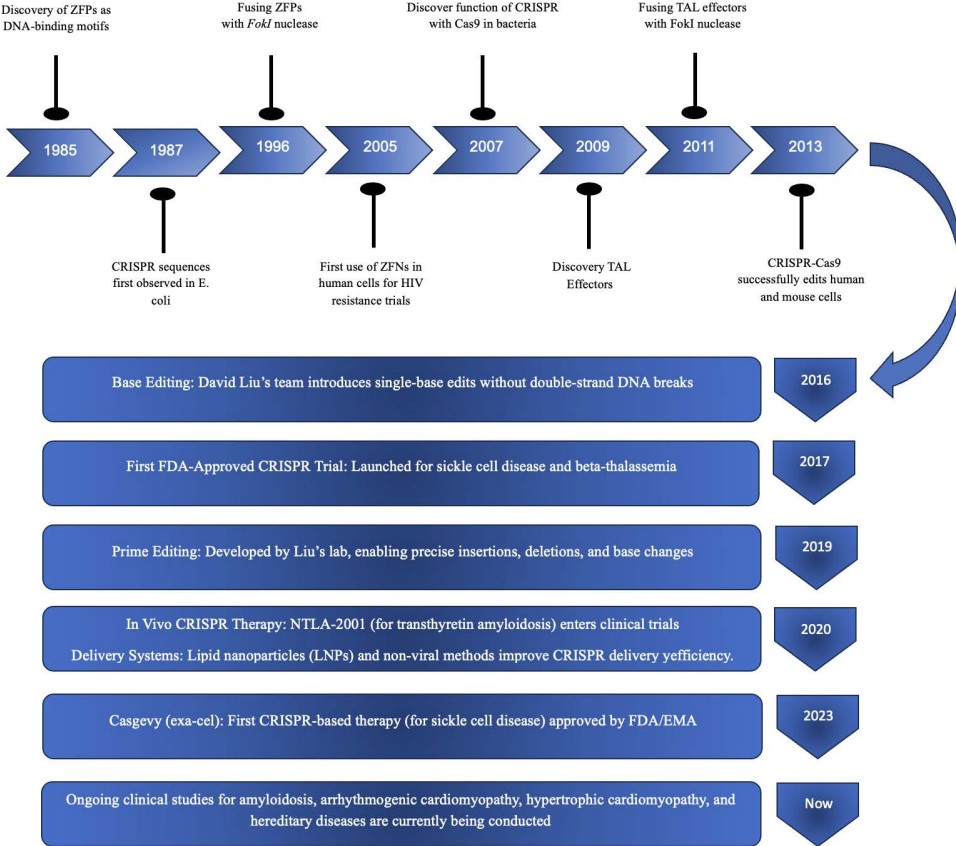

**Fig 3. Milestones of genome editing technology.**

editors (ABEs) – are especially useful for correcting point mutations in monogenic diseases such as hypertrophic cardio-myopathy (HCM). For instance, Reichart et al. (2023) used ABEs to correct the *MYH7* R403Q mutation in an HCM mouse model with over 70% efficiency in cardiomyocytes and minimal side effects [83]. Base editing can also introduce premature stop codons to inactivate genes like PCSK9 [16]. However, these editors are limited to specific transitions (mainly A→G or C→T), and may cause "bystander editing", wherein unintended nearby bases within the editing window are modified, requires careful guide RNA design and thorough validation [15]. Their large size (~5–6 kb) also complicates in vivo delivery, often requiring dual-vector systems [84].

Beyond conventional DNA editing, recent advances in CRISPR-based technologies have expanded to include RNA and epigenome editing, offering powerful tools to modulate gene expression without inducing permanent changes to the genome [85]. These systems typically use catalytically inactive Cas13 (dCas13) fused to enzymatic domains such as ADAR2 or APOBEC for A-to-I or C-to-U editing, guided by crRNAs. Additionally, RNA methylation can be modulated by fusing dCas13 with methyltransferases (e.g., METTL3/METTL14) or demethylases (e.g., ALKBH5) to target m6A modifications [86]. RNA editing systems offer transient and reversible modulation of gene expression, reducing the risk of permanent off-target genetic alterations. This property makes RNA-targeted approaches especially suitable for therapeutic or research applications where temporary gene regulation is preferred [87]. Despite these advantages, several challenges remain. Off-target activity, particularly in systems with broad sequence tolerance, continues to pose a challenge. Editing efficiency is also influenced by the sequence and structural context of the target RNA. Furthermore, efficient delivery of these editing components into primary cells and tissues, particularly in vivo, remains a significant technical hurdle [88].

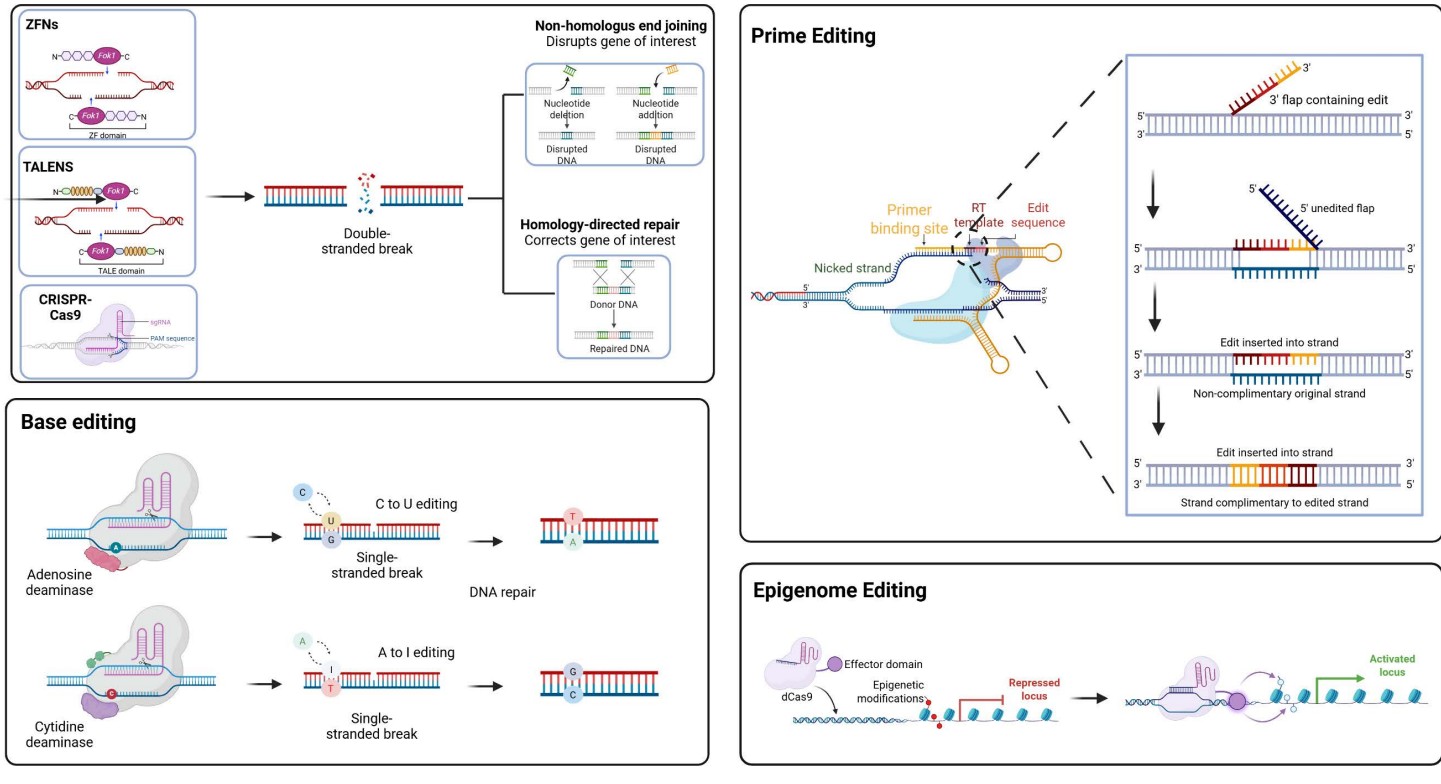

**Fig 4. Gene and epigenome editing strategies.** ZFNs, TALENs, and CRISPR-Cas9 create double-stranded breaks repaired by non-homologous end joining or homology-directed repair. Prime editing uses a Cas9 nickase and reverse transcriptase to introduce precise edits without double-strand breaks. Base editors convert specific nucleotides (e.g., C to T, A to G) using deaminase enzymes. Epigenome editing uses dCas9 fused to effector domains to modulate gene expression without changing the DNA sequence.

Epigenome editing offers a promising approach by using catalytically inactive Cas9 (dCas9) fused to epigenetic modifiers like transcriptional activators (e.g., VP64, p300) or repressors (e.g., KRAB) to modulate gene expression without altering the DNA sequence [89]. This method enables reversible and tunable control through targeted histone or DNA methylation changes. Transient promoter methylation, known as the "hit-and-run" approach, has demonstrated durable therapeutic effects [90]. While it avoids risks linked to DSBs, off-target gene modulation remains a concern, necessitating stringent specificity testing. Furthermore, the delivery of large fusion proteins requires optimized vectors, often using dual-vector systems akin to those in base editing [58]. Epigenome editing shows potential for silencing harmful genes or activating protective pathways, especially in cardiovascular disease models [91] (Fig 4).

## Therapeutic outcomes

Among the included studies, CRISPR-Cas9 was the most frequently employed gene editing platform, predominantly targeting hyperlipidemia-related genes such as *PCSK9*, *LDLR*, and *ANGPTL3*. While CRISPR-Cas9 offers high efficiency and versatility, its reliance on homology-directed repair (HDR) limits its precision, with reported *HDR* efficiencies as low as 6.7% for *LDLR* editing [27]. As an alternative, base editors enable single-nucleotide conversions without inducing double-strand breaks, offering greater precision and reduced risk of off-target effects. In the study by Grosch et al. (2022), base editing systems achieved high editing efficiencies ranging from 70% to 87% [44]. However, their application is restricted to specific types of point mutations and may be less effective in larger animal models. For example, Rothgangl et al. (2021) observed editing efficiencies of 35–40% in macaques, compared to 58% in mice [92]. Less commonly used

platforms, such as CRISPR-CasRx and CRISPR-Cas10, were also reported, highlighting the expanding versatility of CRISPR-based technologies [30,39]. The choice of gene editing tool is determined by various factors, such as the specific target gene selection, the intended therapeutic objective, the type of delivery vector, and the characteristics of the animal experimental model. This highlights the importance of adopting context-specific strategies to ensure optimal outcomes.

## Target gene selection

Gene therapy targets the organ where the disease-driving gene is predominantly expressed or has its pathological effect. In cardiovascular gene editing, the fundamental strategic decision involves whether to direct the gene editing machinery to the heart itself or to a peripheral organ such as the liver to achieve the intended therapeutic outcome.

The liver plays a central role in the production of lipoproteins and enzymes that influence numerous cardiovascular risk factors. Key genes such as PCSK9, ANGPTL3, and TTR are highly expressed in hepatocytes and contribute to systemic blood traits (cholesterol levels, circulating proteins) that have direct effects on cardiovascular health [93]. In a notable example, Richard G Lee et al (2023) employed single LNP-based infusion of the base editor VERVE-101 in non-human primates resulted in an 83% reduction in circulating PCSK9 protein and a ~ 69% decrease in LDL-C, with effects lasting over a year [51]. Similarly, first in vivo CRISPR clinical trial targeted TTR using lipid nanoparticles (LNPs) to deliver Cas9 to the liver, resulting in an 87% reduction in circulating mutant TTR protein [94]. These examples illustrate a clear rationale: when the pathogenic factor originates from the liver, hepatic editing can confer systemic cardiovascular benefits. Moreover, the liver is an accessible and efficient target for gene delivery due to its high perfusion and natural uptake of LNPs and AAV8 vectors, making it particularly suitable for treating metabolic cardiovascular diseases such as hyperlipidemia and amyloidosis through intravenous administration [95].

In contrast, disorders originating within cardiac tissue, such as inherited cardiomyopathies or arrhythmogenic conditions, require direct gene editing within cardiomyocytes or related cardiac cells [96]. For example, correcting pathogenic mutations like those in the *MYH6* gene associated with hypertrophic cardiomyopathy necessitates efficient delivery of gene editing agents specifically to cardiac muscle cells, enabling restoration of normal myocardial function [47]. However, cardiac-targeted gene editing presents distinct delivery challenges when compared to hepatic gene editing. Conventional systemic administration of vectors, such AAVs or LNPs, typically results in predominant hepatic accumulation [97,98]. In addition, the complex, multicellular architecture of cardiac tissue, combined with the low efficiency of HDR in largely post-mitotic cardiomyocytes, presents significant barriers to effective genome editing in the heart [99]. Consequently, efficient myocardial delivery often necessitates the use of specialized approaches, including cardiotropic vector engineering, high-dose systemic administration, or direct intracardiac injection [100].

## Therapeutic objectives

Gene editing strategies in cardiovascular disease differ significantly between rare monogenic disorders and common polygenic conditions in terms of urgency, feasibility, and translational path.

Rare diseases (monogenic), such as hypertrophic cardiomyopathy (e.g., *MYBPC3*, *MYH7* mutations), dilated cardiomyopathy (*LMNA, RBM20*), or inherited arrhythmias (*KCNQ1, RYR2*), are often caused by single, well-characterized mutations [101]. Gene editing offers high curative potential by directly correcting or silencing the causative variant. Proof-of-concept studies in animal models and patient-derived cells have shown promising results, such as ABE-mediated correction of the *MYH7* R403Q mutation preventing HCM in mice [20]. By definition, rare diseases affect a small number of patients; however, they are often prioritized within orphan disease frameworks. Such frameworks can facilitate faster clinical translation owing to the significant unmet medical needs and a regulatory environment typically more accommodating of potential risks, particularly as patients with life-threatening rare diseases may be willing to accept higher risks for potential therapeutic benefits [94]. This initial success has provided important evidence supporting the safety and efficacy of gene editing approaches, thereby facilitating their potential application to a broader range of indications. However, a

major challenge in applying gene editing to rare diseases lies in the requirement for mutation-specific designs, which raises concerns regarding cost-effectiveness and scalability. Although certain recurrent mutations occur in specific populations and may support broader therapeutic applications, many mutations are unique to individual families, limiting the generalizability of treatment strategies [102]. Approaches such as targeting shared exons, allele knockouts, or editing mutation "hotspots" are being explored to improve scalability [103].

In contrast, common cardiovascular diseases – such as atherosclerosis, hypertension, and heart failure – are influenced by multiple genes and environmental factors, making direct editing of all contributors unfeasible [104]. Instead, editing efforts focus on single genes with outsized effects. *PCSK9*, for example, is a validated target; individuals with loss-of-function variants exhibit low LDL levels and reduced cardiovascular risk [105]. Similarly, *LPA* and *ANGPTL3* are being explored for lipid lowering in high-risk patients [106,107]. In the context of common diseases, gene editing must demonstrate clear advantages – such as the potential for a one-time curative treatment – while maintaining an exceptional safety profile, particularly since these interventions may be administered to individuals who are otherwise relatively healthy. Widespread implementation necessitates robust safety data, long-term monitoring, and the development of cost-effective delivery systems [108]. Although these approaches are inherently complex, they hold the potential for substantial public health benefits if key challenges related to safety and scalability can be addressed. Future directions in the field include targeting somatic mutations, such as *TET2* in clonal hematopoiesis of indeterminate potential (CHIP), or introducing protective genetic variants, such as *APOE2*, for disease prevention [109,110]. However, these strategies remain in the early stages of research and development.

## Delivery vector

Effective delivery of gene editing payloads is a critical determinant of therapeutic success, as the ability to reach target cells and tissues directly impacts editing efficiency and clinical translation. The selection of gene delivery vectors is strategically guided by several key factors, including the target tissue, the size of the genetic cargo, and the desired duration of gene editor expression. Currently, three main platforms dominate in vivo delivery strategies: adeno-associated viruses (AAVs), adenoviral vectors (AdVs), and lipid nanoparticles (LNPs), each offering distinct advantages and limitations.

AAVs remain the most widely used vectors for cardiovascular gene editing due to their high transduction efficiency and tissue-specific tropism. AAVs enable sustained transgene expression, which is particularly beneficial in post-mitotic tissues like cardiac muscle. Various serotypes exhibit distinct targeting profiles: AAV1 and AAV6 efficiently transduce skeletal muscle, AAV5 targets airway epithelia and the CNS [97], AAV8 is hepatotropic [111], and AAV9 is notable for its ability to cross the vascular endothelium and transduce cardiomyocytes, skeletal muscle, and neurons [112,113]. However, AAVs are limited by a small packaging capacity (~4.7 kb), restricting the delivery of large editors like full-length Cas9 or base editing systems [114]. Strategies such as dual-AAV delivery or the use of compact Cas variants address this issue. Immune responses pose a significant barrier to AAV-based therapies, primarily due to the high prevalence of pre-existing neutralizing antibodies against common AAV serotypes [115]. To overcome these challenges, engineered capsids (e.g., AAV.MYO, MyoAAV2) and surface modifications (e.g., Gal–TAT peptides) have been developed to improve cardiomyocyte specificity, reduce off-target transduction, and enhance overall editing efficiency while minimizing immunogenicity [116,117]. Recent advances have also explored the use of melittin analogs to enhance endosomal escape and transgene expression; notably, the insertion of the p5RHH peptide into the AAV capsid significantly improved transduction efficiency in vitro and in vivo, including in rAAV-resistant cells and liver tissue, highlighting a promising approach to further optimize AAV-mediated delivery [118].

AdVs were used in 3 included studies for CRISPR-Cas9 delivery, achieving high editing efficiency [18,19,50]. AdVs are characterized by their high cargo capacity – up to ~30 kb in helper-dependent ("gutless") variants – allowing for delivery of large genome editors or entire HDR toolkits within a single vector [119,120]. AdVs also transduce a broad range of dividing and non-dividing cells and do not integrate into the host genome, thereby reducing the risk of insertional

mutagenesis. However, their pronounced immunogenicity severely limits their clinical application for in vivo gene editing. First-generation AdVs have been associated with acute immune responses and hepatotoxicity, and most adults carry pre-existing anti-AdV immunity [121]. Strategies to reduce immunogenicity – such as using rare serotypes or capsid shielding – have had limited success [122]. Consequently, AdVs have seen niche use primarily in large-animal models or preclinical studies where transient, high expression and large cargo delivery are essential.

LNPs have emerged as a leading non-viral delivery platform, particularly for liver-directed gene editing. LNPs encapsulate mRNA or ribonucleoprotein (RNP) complexes and are naturally directed to the liver due to interactions with serum proteins. This property has been exploited for editing liver-specific genes like *PCSK9* and *ANGPTL3* [123]. Musunuru et al. (2021) demonstrated that a single LNP dose achieving about a 60% editing rate of the *PCSK9* gene in primate liver led to a ∼60% reduction in LDL-C, with only a single off-target mutation detected [16]. LNPs are advantageous for their low immunogenicity, enabling repeated administration without eliciting strong immune responses, and their flexible cargo capacity, which can accommodate large mRNA or protein payloads [124].

However, LNPs are generally limited by transient expression, which – while sufficient for permanent genome editing – is a limitation for therapies requiring sustained protein production [125]. Additionally, LNPs typically exhibit strong liver tropism, making extrahepatic delivery (e.g., to the heart or muscle) more challenging [126,127]. Research is now focused on enhancing LNP specificity through targeted modifications, such as N-acetylgalactosamine (GalNAc) conjugation for hepatocyte targeting via the asialoglycoprotein receptor (ASGR), or exploring new formulations capable of efficient delivery to extrahepatic tissues [40,128].

Emerging delivery technologies, such as engineered virus-like particles (eVLPs), represent a novel approach that blends the efficiency of viral vectors with the safety profile of non-viral systems. eVLPs deliver gene editing proteins (e.g., Cas9 or base editors) without viral DNA integration, reducing the risk of insertional mutagenesis. A 2022 study by Banskota et al. showed that intravenous delivery of eVLPs achieved editing in 63% of mouse hepatocytes and reduced hepatic PCSK9 protein by 78%, with minimal off-target effects [53].

## Experimental model

Preclinical testing of cardiovascular gene editing spans small animal models (mice, rats), mid-sized models (rabbits, pigs), and non-human primates (usually macaques), each with their own biology. Results can vary widely between species, and these differences have crucial implications for interpreting data and predicting human outcomes. For instance, Rothgangl et al. (2021) reported that delivery of an adenine base editor via LNPs resulted in approximately 35–40% allele editing in the myocardium of non-human primates, whereas comparable strategies in mouse liver models achieved significantly higher efficiencies (~58%) [92].

Small animal models, such as mice and rats, are widely used due to their low cost, short reproductive cycles, and well-mapped genomes. However, key physiological differences – especially in lipid metabolism and cardiovascular function – limit their translational value. Mice primarily transport cholesterol via HDL rather than LDL and require genetic modifications (e.g., ApoE or LDLR knockouts) to model human-like atherosclerosis [129]. Their high heart rates (~250–500 bpm) and absence of collateral coronary circulation reduce their suitability for ischemia studies [130]. Furthermore, immunological differences are notable, as mice typically lack pre-existing immunity to AAV or Cas9, unlike humans and primates [131].

Mid-sized models, like rabbits and pigs, offer greater physiological relevance. Rabbits develop LDL-driven atherosclerosis on high-cholesterol diets and are well-suited for evaluating lipid-targeting gene therapies [132]. Their size also permits advanced imaging techniques. Pigs closely mimic human cardiovascular anatomy, lipoprotein profiles, and drug metabolism, making them ideal for testing gene therapies for myocardial infarction and arrhythmias. However, gene editing in pigs is technically demanding, often requiring somatic cell nuclear transfer or advanced delivery systems [133].

Non-human primates, particularly macaques, are the most translationally relevant models due to their close genetic and physiological similarity to humans. They accurately replicate human lipid profiles, insulin responses, and cardiovascular

regulation, enabling comprehensive evaluation of gene-editing therapies. Yet, their use is limited by ethical concerns, cost, and long lifespans, making them suitable mainly for late-stage preclinical studies where safety and immune responses are critical [134].

In conclusion, while rodent models support early-stage research, translational success requires complementary data from larger animals. Differences in physiology and immunity across species remain key considerations, and regulatory approval often necessitates safety data from NHPs before initiating human trials.

### Safety and off-target effects

Safety assessments across the included studies consistently identified transient elevations in liver enzymes as the most commonly observed adverse effect. Notably, off-target gene editing events were minimal, occurring at a frequency of less than 1% in the majority of studies. No life-threatening adverse events were reported, suggesting favorable short-term safety of gene editing interventions in cardiovascular disease model. However, long-term safety remains insufficiently characterized. Key concerns such as immunogenicity following repeated vector administration or the potential for onco-genic transformations arising from off-target genomic alterations highlight the need for extended follow-up, especially since most preclinical studies to date have monitored outcomes for only a few weeks to months post-intervention.

A comprehensive evaluation of safety is critical when translating gene editing therapies to clinical settings, as each component of the therapeutic platform can introduce distinct risks. For instance, the use of bacterial-derived nucleases like Cas9 or Cas12 could trigger immune responses in vivo, both animals and humans may harbor pre-existing antibodies against these proteins. [115,135]. High systemic doses of AAV vectors have been associated with immune-mediated hep-atitis in some preclinical and clinical studies, while AdV vectors are well known to induce dose-dependent inflammatory responses [136,137]. LNP-based delivery systems are generally well tolerated, they can trigger transient infusion-related reactions or activate innate immune pathways [126,127].

Editor-specific off-target effects remain a significant concern in the field of gene editing. The CRISPR-Cas9 system, for instance, can occasionally induce unintended double-strand breaks at genomic loci bearing partial sequence homology, potentially resulting in mosaic or undesirable mutations [138]. Likewise, base editors may catalyze bystander nucleotide conversions and have been shown to alter RNA transcripts, thereby introducing unintended genetic modifications [139]. Furthermore, disruption of genes with uncharacterized or broad systemic functions may result in unforeseen phenotypic consequences, while partial correction – where only a subset of target cells is edited – may be insufficient for therapeutic efficacy and could introduce heterogeneity among cell populations [140].

To address these challenges, ongoing innovations are being developed to improve both the safety and specificity of gene editing strategies. Tissue-specific nanoparticles and non-viral delivery methods – such as exosomes and extra-cellular vesicles – are being investigated as alternatives to conventional viral vectors. MyoAAV, a muscle-specific AAV variant, has shown promise in enhancing targeted gene delivery to cardiac and skeletal muscle tissues, [141]. Meanwhile, next-generation editing tools such as prime editing and high-specificity CRISPR variants (e.g., Cas12, Cas13, Cas14) offer improved precision [142,143]. High-fidelity Cas9 variants, including eSpCas9 and HiFi Cas9, have also demonstrated reduced off-target activity, thereby enhancing safety profiles [144]. Finally, the integration of artificial intelligence and machine learning in gRNA design could enhance targeting precision and minimize off-target effects [145,146].

### Future directions: From bench to bedside

Promising outcomes from preclinical models have laid the groundwork for gene editing therapies in human cardiovascular disease. Currently, several clinical trials are underway to evaluate these approaches, particularly those targeting *PCSK9* for the treatment of hypercholesterolemia. Among the most advanced candidates is Verve Therapeutics' VERVE-101, a CRISPR-based therapy targeting the *PCSK9* gene, now in Phase 1 clinical trial for familial hypercholesterolemia. Pre-liminary data indicate substantial reductions in LDL-c levels without significant adverse effects [147]. A related candidate,

VERVE-102, employs the same genetic payload as VERVE-101 but utilizes a GalNAc LNP delivery system; it is currently being assessed in the ongoing Phase 1b HEART-2 trial [148]. In addition, Verve is developing VERVE-201, which targets *ANGPTL3*, offering an alternative therapeutic strategy focused on remnant cholesterol metabolism in patients with refractory hypercholesterolemia. This candidate is also in Phase 1b clinical evaluation [82]. The recent approval by the U.S. Food and Drug Administration (FDA) of Verve's Investigational New Drug (IND) application has facilitated the expansion of clinical trials within the United States. CRISPR-based therapies utilizing LNP delivery systems have similarly shown substantial promise. AccurEdit Therapeutics' ART002, for instance, achieved a 50–70% reduction in LDL-c levels in Phase 1 studies, potentially offering superior durability and efficacy compared to RNA interference (RNAi)-based treatments. These single-dose interventions may also overcome long-standing challenges related to patient adherence in lipid-lowering therapies [149].

In the context of cardiac amyloidosis, NTLA-2001 (Nexiguran ziclumeran or nex-z), a CRISPR-Cas9-based gene-editing therapy, demonstrated an 89% reduction in serum transthyretin levels at 28 days and a 90% reduction at 12 months during Phase 1 trials. Clinical outcomes indicated stability or improvement in 92% of patients, accompanied by stable NT-proBNP and troponin levels, and a favorable safety profile characterized by only mild adverse events [150]. Furthermore, Intellia Therapeutics' NTLA-2001, delivered via LNPs, achieved over 80% TTR knockdown and was associated with improvements in cardiac performance. The therapy has been well tolerated in Phase 2 study, and the ongoing Phase 3 MAGNITUDE trial (NCT06128629) is expected to conclude in 2028 [151]. For inherited cardiomyopathies, AAV-based gene delivery approaches have shown early potential. Tenaya Therapeutics' TN-201, designed to target *MYBPC3* in hypertrophic cardiomyopathy, has demonstrated successful vector delivery and transgene expression in cardiac tissue [152]. Similarly, Regenxbio's RGX-202, initially developed for Duchenne muscular dystrophy, has shown promising micro-dystrophin expression and may offer therapeutic benefits for the associated cardiomyopathy [153] (Table 5).

## Ethical considerations

The clinical implementation of gene editing therapies for CVDs must be guided by careful ethical oversight and comprehensive regulatory frameworks. As these interventions permanently alter the genome, there is a ethical obligation to ensure that potential benefits justify the risks and that unintended consequences are minimized. A central ethical imperative is to maintain public trust. To this end, the medical and scientific communities must maintain transparency regarding both the known risks – such as off-target effects and immune responses – and the uncertainties, including potential long-term outcomes. Active and ongoing engagement with patients and the broader public is essential to foster informed understanding and acceptance of these emerging therapies [155].

Public sensitivity to gene editing has been heightened by controversial events, such as the unethical germline genome editing of human embryos reported in 2018, which underscored the need to clearly distinguish therapeutic somatic gene editing from heritable germline modifications. International consensus statements, including the National Academy of Sciences 2017 report, have affirmed that somatic gene editing may be ethically permissible under stringent regulatory oversight, whereas germline editing remains widely prohibited across jurisdictions [156]. Researchers and clinicians bear the responsibility to adhere to established ethical guidelines and regulatory standards. This includes obtaining comprehensive informed consent, which must explicitly communicate the irreversible nature of gene editing procedures and the necessity for long-term, potentially lifelong, clinical monitoring of individuals who receive such therapies.

Equitable access and justice considerations also arise as gene editing therapies move toward clinical application. Perhaps the most pressing challenges is the cost. For example, the recently approved gene therapy for sickle cell disease, Casgevy, has a reported price of $2 million, raising serious questions about equitable access and reimbursement. This is especially concerning for chronic conditions such as CVDs, which affect large, diverse patient populations. It is a moral imperative to ensure that life-saving gene editing interventions are not limited to a socioeconomically privileged minority. Policies and funding mechanisms must be developed to promote broader accessibility and prevent the exacerbation of existing healthcare disparities.

**Table 5. Ongoing clinical trials in gene therapy of CVD. From the American Heart Association Advisory [154].**

| Trial ID | Start Year | Disease/ Condition | Study Design | Target Gene/ Protein | Therapy Name | Delivery Method | Current Trial Phase |
|---|---|---|---|---|---|---|---|
| **Genetic Cardiomyopathy** | | | | | | | |
| NCT05885412 | 2023 | Arrhythmogenic Cardiomyopathy (ACM) | Open Label Multi center | PKP2 | RP-A601 | AAV | Phase 1 |
| NCT06109181 | 2024 | ACM | Open Label Multi center | PKP2 | LX2020 | AAV | Phase 1/2 |
| NCT06228924 | 2024 | ACM | Open Label Multi center | PKP2 | TN-401 | AAV | Phase 1 |
| NCT05836259 | 2023 | Hypertrophic Cardiomyopathy (HCM) | Open Label Multi center | MYBPC3 | TN-201 | AAV | Phase 1b |
| **Heart Failure** | | | | | | | |
| NCT05598333 | 2023 | Ischemic Cardiomyopathy & Heart Failure | Double-blinded Multi center | PP1 | AB-1002 | AAV | Phase 2 |
| **Hyperlipidemia** | | | | | | | |
| NCT06125847 | 2023 | FH | Open-label Single-center | LDLR | NGGT006 | AAV | Phase 1 |
| NCT00891306 | 2009 | FH | Open Label Multi center | LDLR | LPLS447X | AAV | Phase 2/3 |
| NCT06293729 | 2024 | FH | Open-label Single-center | LDLR | NGGT006 | AAV | Phase 1 |
| NCT06112327 | 2024 | FH | Long-term follow-up | PCSK9 | VERVE-101 | LNP | Phase 1 |
| NCT05860569 | 2024 | Hypertriglyceridemia | Open Label Multi center | LPL | GC304 | AAV | Phase 1 |
| **Genetic Syndromes** | | | | | | | |
| NCT04601051 | 2020 | Transthyretin Amyloidosis | Open Label Multi center | TTR | NTLA-2001 | LNP | Phase 1 |
| NCT05445323 | 2022 | Friedreich's Ataxia | Open Label Multi center | FXN | LX2006 | AAV | Phase 1/2 |
| NCT05302271 | 2022 | Friedreich's Ataxia | Open-label Single-center | FXN | AAVrh.10hFXN | AAV | Phase 1a |
| NCT04174105 | 2020 | Pompe Disease | Open Label Multi center | GAA | AT845 | AAV | Phase 1 |
| NCT03533673 | 2018 | Pompe Disease | Open-label Single-center | GAA | ACTUS-101 | AAV | Phase 1/2 |
| NCT04093349 | 2020 | Pompe Disease | Open Label Multi center | GAA | SPK-3006 | AAV | Phase 1/2 |
| NCT00976352 | 2010 | Pompe Disease | Open-label Single-center | GAA | AAV-GAA | AAV | Phase 1/2 |
| NCT02240407 | 2017 | Pompe Disease | Double-blinded Single-center | GAA | AAV-GAA | AAV | Phase 1 |
| NCT04046224 | 2019 | Fabry Disease | Open Label Multi center | GLA | ST-920 | AAV | Phase 1/2 |
| NCT04519749 | 2020 | Fabry Disease | Open Label Multi center | GLA | 4D-310 | AAV | Phase 1/2 |
| NCT06092034 | 2023 | Danon Disease | Open Label Multi center | LAMP2 | RP-A501 | AAV | Phase 2 |

Regulatory authorities will also need to address how to ethically design early human trials – for example, selecting appropriate patients (those with no remaining conventional treatment options) and balancing risk-benefit in life-threatening versus less severe conditions. Furthermore, the long-term safety and durability of gene editing therapies remain incompletely understood and warrant sustained investigation [157]. While preclinical studies and early-phase trials have shown promising results, the potential for delayed adverse effects, such as immune responses or off-target mutations,

necessitates long-term monitoring of patients [158]. The establishment of long-term patient registries will likely be essential to track clinical outcomes and identify any late-emerging complications, ensuring continuous data collection beyond the confines of initial trials. Moreover, as with any transformative biomedical technology, the ethical governance of gene editing must be dynamic and responsive. This includes ongoing ethical review processes, periodic reassessment of regulatory frameworks in light of emerging scientific evidence or evolving societal values, and active involvement of diverse stakeholders – including ethicists, patient advocacy groups, clinicians, and policymakers – in guiding the responsible development of these therapies.

## Conclusion

This study highlights the tremendous potential of gene editing therapies for treating CVDs. The high editing efficiencies, significant therapeutic outcomes, and favorable safety profiles observed in animal models provide a strong rationale for advancing these therapies to clinical trials. Notably, unlike conventional drugs or interventions that require ongoing treatment to control disease symptoms, gene editing offers the possibility of a one-time, curative intervention by directly correcting the underlying genetic defect. However, challenges related to delivery, long-term safety, and scalability must be addressed to fully realize the potential of gene editing in cardiovascular medicine. With continued innovation and rigorous preclinical and clinical evaluation, gene editing therapies could revolutionize the treatment of CVDs, offering hope for patients with currently untreatable conditions.

## Supporting information

**S1 File. Supplemental materials.**
(DOCX)

**S2 File. PRISMA checklist.**
(DOCX)

## Acknowledgments

Fig 4 was created using BioRender (https://www.biorender.com).

## Author contributions

**Conceptualization:** Quan Duy Vo.

**Data curation:** Quan Duy Vo.

**Formal analysis:** Quan Duy Vo.

**Investigation:** Quan Duy Vo.

**Project administration:** Quan Duy Vo.

**Resources:** Quan Duy Vo.

**Software:** Quan Duy Vo.

**Supervision:** Quan Duy Vo.

**Validation:** Quan Duy Vo.

**Visualization:** Quan Duy Vo.

**Writing – original draft:** Quan Duy Vo.

**Writing – review & editing:** Quan Duy Vo.

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
