## [Decision Letter · Decision Letter 0]

28 Mar 2025

Dear Dr. Vo,

Thank you for submitting your manuscript to PLOS ONE. After careful consideration, we feel that it has merit but does not fully meet PLOS ONE’s publication criteria as it currently stands. Therefore, we invite you to submit a revised version of the manuscript that addresses the points raised during the review process.

We look forward to receiving your revised manuscript.

Kind regards,

Chen Ling, Ph.D.

Academic Editor

PLOS ONE

Additional Editor Comments (if provided):

Reviewers' comments:

Reviewer's Responses to Questions

**Comments to the Author**

1. Is the manuscript technically sound, and do the data support the conclusions?

Reviewer #1: Yes

Reviewer #2: Partly

Reviewer #3: Yes

2. Has the statistical analysis been performed appropriately and rigorously?

Reviewer #1: N/A

Reviewer #2: N/A

Reviewer #3: Yes

3. Have the authors made all data underlying the findings in their manuscript fully available?

Reviewer #1: Yes

Reviewer #2: Yes

Reviewer #3: Yes

4. Is the manuscript presented in an intelligible fashion and written in standard English?

Reviewer #1: Yes

Reviewer #2: Yes

Reviewer #3: Yes

Reviewer #1: 1. The author should add the full name before the first use of a certain abbreviation�including “CRISPR-Cas9”�. And the Introduction part should include general introduction�history, background�, why to write this review, the focus and highlights of this review, the following logical sequence of the review, best presented in 3-5 paragraphs.

2. Considering that most of the included studies used the CRISPR system, I suggest the author add a separate section to summarize the discovery and development of the basic CRISPR system and base editors.

3. I understand that the authors‘ main focus is on screened articles, but the authors mention the use of gene editing therapies in animal models in the title. I would suggest that the authors add a separate section to discuss the impact of various types of animal models in gene editing therapies for cardiovascular disease, especially given that some studies are currently using rabbits, dogs and cats, among others.

4. Ethical issues of gene editing treatments should be mentioned.

5. AAV subtypes (AAV8, AAV9) should be discussed.

Reviewer #2: I like the topic of this review. If it could be properly written, it will be an important reference for the field. However, there are major problems in the contents and logics. The following questions need to be solved:

1. There must be something wrong with keyword search. You need to truly understand this field in order to write a good review paper. You need to curate literature search manually since some good papers could not be easily searched by just looking at title or abstract. Just to give you some examples of missed key papers in high-profile journals:

Lebek S, Chemello F, Caravia XM, Tan W, Li H, Chen K, Xu L, Liu N, Bassel-Duby R, Olson EN. Ablation of CaMKIIδ oxidation by CRISPR-Cas9 base editing as a therapy for cardiac disease. Science. 2023 Jan 13;379(6628):179-185. doi: 10.1126/science.ade1105. Epub 2023 Jan 12. PMID: 36634166; PMCID: PMC10150399.

Lebek S, Caravia XM, Chemello F, Tan W, McAnally JR, Chen K, Xu L, Liu N, Bassel-Duby R, Olson EN. Elimination of CaMKIIδ Autophosphorylation by CRISPR-Cas9 Base Editing Improves Survival and Cardiac Function in Heart Failure in Mice. Circulation. 2023 Nov 7;148(19):1490-1504. doi: 10.1161/CIRCULATIONAHA.123.065117. Epub 2023 Sep 15. PMID: 37712250; PMCID: PMC10842988.

Wu S, Yang P, Geng Z, Li Y, Guo Z, Lou Y, Zhang S, Xiong J, Hu H, Guo X, Pu WT, Zhang Y, Zhu D, Zhang B. Base editing effectively prevents early-onset severe cardiomyopathy in Mybpc3 mutant mice. Cell Res. 2024 Apr;34(4):327-330. doi: 10.1038/s41422-024-00930-7. Epub 2024 Feb 9. PMID: 38337022; PMCID: PMC10978934.

Liu Z, Yang L, Yang Y, Li J, Chen Z, Guo C, Guo Q, Li Q, Zhao D, Hu X, Gao F, Guo Y. ABE-Mediated Cardiac Gene Silencing via Single AAVs Requires DNA Accessibility. Circ Res. 2025 Jan 31;136(3):318-320. doi: 10.1161/CIRCRESAHA.124.325611. Epub 2025 Jan 16. PMID: 39817340.

Yang L, Liu Z, Chen G, Chen Z, Guo C, Ji X, Cui Q, Sun Y, Hu X, Zheng Y, Li Y, Gao F, Chen L, Zhou P, Pu WT, Guo Y. MicroRNA-122-Mediated Liver Detargeting Enhances the Tissue Specificity of Cardiac Genome Editing. Circulation. 2024 May 28;149(22):1778-1781. doi: 10.1161/CIRCULATIONAHA.123.065438. Epub 2024 May 28. PMID: 38805581.

Ghahremani S, Kanwal A, Pettinato A, Ladha F, Legere N, Thakar K, Zhu Y, Tjong H, Wilderman A, Stump WT, Greenberg L, Greenberg MJ, Cotney J, Wei CL, Hinson JT. CRISPR Activation Reverses Haploinsufficiency and Functional Deficits Caused by TTN Truncation Variants. Circulation. 2024 Apr 16;149(16):1285-1297. doi: 10.1161/CIRCULATIONAHA.123.063972. Epub 2024 Jan 18. PMID: 38235591; PMCID: PMC11031707.

Ma S, Jiang W, Liu X, Lu WJ, Qi T, Wei J, Wu F, Chang Y, Zhang S, Song Y, Bai R, Wang J, Lee AS, Zhang H, Wang Y, Lan F. Efficient Correction of a Hypertrophic Cardiomyopathy Mutation by ABEmax-NG. Circ Res. 2021 Oct 29;129(10):895-908. doi: 10.1161/CIRCRESAHA.120.318674. Epub 2021 Sep 16. PMID: 34525843.

2. the paper need to be rewriten to discuss some key concepts with better structures and logics:

AAV vs LNP vs AdV: what are differences between these vectors? The Pros and Cons of each of these vectors should be analysed and described. Why some studies used AAV while other studies used LNP? What is the rationale behind vector choices?

Heart vs Liver: why some studies target the liver while other studies target the heart? Are they treating different type of diseases? What is the rationale behind organ choices? Are there difficulties or concerns to deliver to each organ?

Cas9 NHEJ vs base editing vs epigenome editing vs RNA editing vs HDR: the paper should explain working mechanisms of each editor and their strength or weakness. Why certain studies used certain editor?

Rare disease versus common disease: some therapies are specific for rare disease only (such as ABE correction of rare disease mutation). Other therapies are suitable for both rare and common diseases (such as pcsk9 and camk2d studies). Analysis from this angle is necessary to determine which type of therapy is more realistic and promising.

Safety concerns from vector, editor, gene target should be separately and more detailed and systemically summarized.

Reviewer #3: This review examines the application of gene editing technologies in animal models of CVDs, summarizing the therapeutic efficacy and safety profiles of tools such as CRISPR-Cas9 in treating hyperlipidemia, cardiomyopathy, and related conditions. The article also discusses current limitations and future directions, including clinical translation and ethical considerations. While the manuscript presents a logically structured and comprehensive overview, certain aspects require supplementation.

1. The Introduction fails to explicitly delineate the fundamental distinctions between gene editing and conventional therapies (e.g., target specificity). The latest studies such as PMID: 37662968 can be referenced.

2. Page 8 states "Lipid nanoparticles were employed...to the liver (PCSK9, LDLR)" without explaining why the liver serves as a critical therapeutic target for CVD treatment, creating an incomplete logical progression.

3. Page 8 notes "editing efficiency varies across species" but lacks mechanistic analysis of underlying causes.

4. Page 14 could benefit from incorporation of recent findings (e.g., PMID: 40061823).

5. Page 18 requires more detailed technical descriptions of gene editing protocols, particularly CRISPR-Cas9 sgRNA design principles.

6. Page 11 should address interspecies biological differences in gene editing responses (e.g., mice vs. non-human primates) and their clinical translation implications.

7. Pages 20-23 would be strengthened by including updates on human clinical trials, particularly late-stage or approved therapies.

8. The verbose sentence on Page 4 ("Early gene replacement therapies...") could be condensed to: "Early therapies introduced functional gene copies to restore normal function."

9. In Introduction or Discussion, current advances in AAV research (e.g., transduction efficiency enhancement) should be cited (e.g., PMID: 38307819).

10. Terminology inconsistency ("CRISPR-Cas9" vs. "CRISPR/Cas9") requires standardization.

**Do you want your identity to be public for this peer review?** For information about this choice, including consent withdrawal, please see our Privacy Policy

Reviewer #1: No

Reviewer #2: **Yes: ** Yuxuan Guo

Reviewer #3: No

---

## [Author Response · Author response to Decision Letter 1]

7 Apr 2025

I would like to express my gratitude for your thoughtful and constructive feedback on my manuscript. Your detailed comments have been invaluable in guiding the revision process. In response to your suggestions, I have meticulously addressed each point to enhance the clarity, completeness, and precision of my work. Below, I outline the revisions made to the manuscript based on your insightful remarks.

Reviewer #1:

1. The author should add the full name before the first use of a certain abbreviation including “CRISPR-Cas9”. And the Introduction part should include general introduction history, background, why to write this review, the focus and highlights of this review, the following logical sequence of the review, best presented in 3-5 paragraphs

Respond:

Thank you for the suggestion. I have provided the full name for all abbreviations upon their first use. Additionally, I have revised the Introduction to include the historical context, the rationale for conducting this review, the key highlights, and the logical structure, as recommended.

2. Considering that most of the included studies used the CRISPR system, I suggest the author add a separate section to summarize the discovery and development of the basic CRISPR system and base editors.

Respond:

I appreciate this recommendation. A dedicated section summarizing the history and development of the CRISPR system and base editors has been added to the Discussion section.

3. I understand that the authors‘ main focus is on screened articles, but the authors mention the use of gene editing therapies in animal models in the title. I would suggest that the authors add a separate section to discuss the impact of various types of animal models in gene editing therapies for cardiovascular disease, especially given that some studies are currently using rabbits, dogs and cats, among others

Respond:

In response to this valuable suggestion, I have added a dedicated section that discusses the roles of various animal species used in gene editing studies for cardiovascular disease, including their respective advantages and translational significance.

4. Ethical issues of gene editing treatments should be mentioned.

Respond:

I have expanded the discussion on ethical considerations to specifically address the implications, challenges, and regulatory concerns associated with gene editing technologies in preclinical and clinical contexts.

5. AAV subtypes (AAV8, AAV9) should be discussed.

Respond:

A comparative analysis of AAV subtypes, including AAV8 and AAV9, has been incorporated into the section titled “Delivery Vector Comparisons” within the Discussion, highlighting their tissue tropism, efficacy, and potential clinical applications.

Reviewer #2:

1. There must be something wrong with keyword search. You need to truly understand this field in order to write a good review paper. You need to curate literature search manually since some good papers could not be easily searched by just looking at title or abstract. Just to give you some examples of missed key papers in high-profile journals:

• Lebek S, Chemello F, Caravia XM, Tan W, Li H, Chen K, Xu L, Liu N, Bassel-Duby R, Olson EN. Ablation of CaMKIIδ oxidation by CRISPR-Cas9 base editing as a therapy for cardiac disease. Science. 2023 Jan 13;379(6628):179-185. doi: 10.1126/science.ade1105. Epub 2023 Jan 12. PMID: 36634166; PMCID: PMC10150399.

• Lebek S, Caravia XM, Chemello F, Tan W, McAnally JR, Chen K, Xu L, Liu N, Bassel-Duby R, Olson EN. Elimination of CaMKIIδ Autophosphorylation by CRISPR-Cas9 Base Editing Improves Survival and Cardiac Function in Heart Failure in Mice. Circulation. 2023 Nov 7;148(19):1490-1504. doi: 10.1161/CIRCULATIONAHA.123.065117. Epub 2023 Sep 15. PMID: 37712250; PMCID: PMC10842988.

• Wu S, Yang P, Geng Z, Li Y, Guo Z, Lou Y, Zhang S, Xiong J, Hu H, Guo X, Pu WT, Zhang Y, Zhu D, Zhang B. Base editing effectively prevents early-onset severe cardiomyopathy in Mybpc3 mutant mice. Cell Res. 2024 Apr;34(4):327-330. doi: 10.1038/s41422-024-00930-7. Epub 2024 Feb 9. PMID: 38337022; PMCID: PMC10978934.

• Liu Z, Yang L, Yang Y, Li J, Chen Z, Guo C, Guo Q, Li Q, Zhao D, Hu X, Gao F, Guo Y. ABE-Mediated Cardiac Gene Silencing via Single AAVs Requires DNA Accessibility. Circ Res. 2025 Jan 31;136(3):318-320. doi: 10.1161/CIRCRESAHA.124.325611. Epub 2025 Jan 16. PMID: 39817340.

• Yang L, Liu Z, Chen G, Chen Z, Guo C, Ji X, Cui Q, Sun Y, Hu X, Zheng Y, Li Y, Gao F, Chen L, Zhou P, Pu WT, Guo Y. MicroRNA-122-Mediated Liver Detargeting Enhances the Tissue Specificity of Cardiac Genome Editing. Circulation. 2024 May 28;149(22):1778-1781. doi: 10.1161/CIRCULATIONAHA.123.065438. Epub 2024 May 28. PMID: 38805581

• Ghahremani S, Kanwal A, Pettinato A, Ladha F, Legere N, Thakar K, Zhu Y, Tjong H, Wilderman A, Stump WT, Greenberg L, Greenberg MJ, Cotney J, Wei CL, Hinson JT. CRISPR Activation Reverses Haploinsufficiency and Functional Deficits Caused by TTN Truncation Variants. Circulation. 2024 Apr 16;149(16):1285-1297. doi: 10.1161/CIRCULATIONAHA.123.063972. Epub 2024 Jan 18. PMID: 38235591; PMCID: PMC11031707.

• Ma S, Jiang W, Liu X, Lu WJ, Qi T, Wei J, Wu F, Chang Y, Zhang S, Song Y, Bai R, Wang J, Lee AS, Zhang H, Wang Y, Lan F. Efficient Correction of a Hypertrophic Cardiomyopathy Mutation by ABEmax-NG. Circ Res. 2021 Oct 29;129(10):895-908. doi: 10.1161/CIRCRESAHA.120.318674. Epub 2021 Sep 16. PMID: 34525843.

Respond:

Thank you for highlighting these omissions and providing valuable references. I would like to clarify that some of the suggested articles (e.g., doi: 10.1038/s41422-024-00930-7, 10.1161/CIRCRESAHA.124.325611, and 10.1161/CIRCULATIONAHA.123.065438) were identified during my initial search but were excluded as they are letters or correspondences, which did not meet our inclusion criteria focused on original research. Additionally, the article with doi: 10.1161/CIRCULATIONAHA.123.063972 was excluded due to its use of a cell-based model rather than an animal model. However, I acknowledge that three important articles (doi: 10.1126/science.ade1105, 10.1161/CIRCULATIONAHA.123.065117, and 10.1161/CIRCRESAHA.120.318674) were inadvertently omitted. I have now revised the search strategy and manually curated additional studies, which has increased the total number of included studies to 50. However, I acknowledge that even with the modified search query, some articles may not have been included in the final analysis.Furthermore, the inclusion criteria have been clarified for better transparency.

2. the paper need to be rewriten to discuss some key concepts with better structures and logics:

AAV vs LNP vs AdV: what are differences between these vectors? The Pros and Cons of each of these vectors should be analysed and described. Why some studies used AAV while other studies used LNP? What is the rationale behind vector choices?

Heart vs Liver: why some studies target the liver while other studies target the heart? Are they treating different type of diseases? What is the rationale behind organ choices? Are there difficulties or concerns to deliver to each organ?

Cas9 NHEJ vs base editing vs epigenome editing vs RNA editing vs HDR: the paper should explain working mechanisms of each editor and their strength or weakness. Why certain studies used certain editor?

Rare disease versus common disease: some therapies are specific for rare disease only (such as ABE correction of rare disease mutation). Other therapies are suitable for both rare and common diseases (such as pcsk9 and camk2d studies). Analysis from this angle is necessary to determine which type of therapy is more realistic and promising.

Safety concerns from vector, editor, gene target should be separately and more detailed and systemically summarized.

Respond:

Thank you for the insightful feedback. I have substantially revised the Discussion section to include dedicated subsections addressing each of these important topics.

Reviewer #3:

1. The Introduction fails to explicitly delineate the fundamental distinctions between gene editing and conventional therapies (e.g., target specificity). The latest studies such as PMID: 37662968 can be referenced.

Respond:

I have revised the Introduction to more clearly highlight the mechanistic and therapeutic distinctions between gene editing approaches and conventional treatments. The suggested reference (PMID: 37662968) has been cited to support this revision.

2. Page 8 states "Lipid nanoparticles were employed...to the liver (PCSK9, LDLR)" without explaining why the liver serves as a critical therapeutic target for CVD treatment, creating an incomplete logical progression.

Respond:

To enhance the logical coherence of the manuscript, a new subsection has been added to the Discussion to explain the rationale for targeting the liver in cardiovascular therapy, including a summary of key metabolic and regulatory roles of hepatic tissue.

3. Page 8 notes "editing efficiency varies across species" but lacks mechanistic analysis of underlying causes.

Respond:

I have expanded the “Species Models” section to include a mechanistic discussion on factors influencing interspecies variability in gene editing outcomes, including genomic context, immune response, and delivery efficiency..

4. Page 14 could benefit from incorporation of recent findings (e.g., PMID: 40061823).

Respond:

The recommended study (PMID: 40061823) has been reviewed and cited in the updated manuscript to support the discussion of recent advances.

5. Page 18 requires more detailed technical descriptions of gene editing protocols, particularly CRISPR-Cas9 sgRNA design principles.

Respond:

A technical explanation of CRISPR-Cas9 sgRNA design and base editing strategies has been added to the section “The development of gene editing systems” in the Discussion sectiom.

6. Page 11 should address interspecies biological differences in gene editing responses (e.g., mice vs. non-human primates) and their clinical translation implications.

Respond:

This important point has been addressed by expanding the “Species Models” section to discuss interspecies differences, such as those between mice and non-human primates, and their implications for translational research.

7. Pages 20-23 would be strengthened by including updates on human clinical trials, particularly late-stage or approved therapies.

Respond:

The manuscript has been updated to include the latest developments in human clinical trials for gene editing-based therapies, with an emphasis on ongoing and late-stage studies relevant to cardiovascular applications.

8. The verbose sentence on Page 4 ("Early gene replacement therapies...") could be condensed to: "Early therapies introduced functional gene copies to restore normal function."

Respond:

The sentence has been revised for conciseness as follows: “Early therapies introduced functional gene copies to restore normal function.”

9. In Introduction or Discussion, current advances in AAV research (e.g., transduction efficiency enhancement) should be cited (e.g., PMID: 38307819).

Respond:

Recent advances in AAV vector development, particularly with respect to enhanced tissue transduction efficiency, have been included in the Discussion. The suggested reference (PMID: 38307819) has also been cited.

10. Terminology inconsistency ("CRISPR-Cas9" vs. "CRISPR/Cas9") requires standardization.

Respond:

I have reviewed the manuscript and standardized all terminology to “CRISPR-Cas9” for consistency and clarity.

---

## [Decision Letter · Decision Letter 1]

30 Apr 2025

Dear Dr. Vo,

Thank you for submitting your manuscript to PLOS ONE. After careful consideration, we feel that it has merit but does not fully meet PLOS ONE’s publication criteria as it currently stands. Therefore, we invite you to submit a revised version of the manuscript that addresses the points raised during the review process.

We look forward to receiving your revised manuscript.

Kind regards,

Chen Ling, Ph.D.

Academic Editor

PLOS ONE

Reviewers' comments:

Reviewer's Responses to Questions

**Comments to the Author**

Reviewer #2: (No Response)

Reviewer #3: All comments have been addressed

2. Is the manuscript technically sound, and do the data support the conclusions?

Reviewer #2: Yes

Reviewer #3: Yes

3. Has the statistical analysis been performed appropriately and rigorously?

Reviewer #2: N/A

Reviewer #3: Yes

4. Have the authors made all data underlying the findings in their manuscript fully available?

Reviewer #2: Yes

Reviewer #3: Yes

5. Is the manuscript presented in an intelligible fashion and written in standard English?

Reviewer #2: Yes

Reviewer #3: Yes

Reviewer #2: The authors addressed most of my questions properly, except for the exclusion of important short-format papers that are still quite important to the field. Research letters are still original research articles, not simple correspondences.

1. Thank you for highlighting these omissions and providing valuable references. I would like to clarify that some of the suggested articles (e.g., doi: 10.1038/s41422-024-00930-7, 10.1161/CIRCRESAHA.124.325611, and 10.1161/CIRCULATIONAHA.123.065438) were identified during my initial search but were excluded as they are letters or correspondences, which did not meet our inclusion criteria focused on original research.

Re: This is exactly the problem with your exclusion criteria. These excluded research letter papers are still original research and they are peer-reviewed following the same standard as research articles. Their shorter formats simply mean they are focused on single strong points that do not need longer formats. You should still include these papers in your review.

Reviewer #3: The author has carefully revised the manuscript according to the comments of the reviewer, and there is no problem now. It is recommended to accept.

**Do you want your identity to be public for this peer review?** For information about this choice, including consent withdrawal, please see our Privacy Policy

Reviewer #2: **Yes: ** Yuxuan Guo

Reviewer #3: No

---

## [Author Response · Author response to Decision Letter 2]

2 May 2025

I would like to express my gratitude for your thoughtful and constructive feedback on my manuscript. Below, I outline the revisions made to the manuscript based on your insightful remarks.

Reviewer #2:

The authors addressed most of my questions properly, except for the exclusion of important short-format papers that are still quite important to the field. Research letters are still original research articles, not simple correspondences.

1. Thank you for highlighting these omissions and providing valuable references. I would like to clarify that some of the suggested articles (e.g., doi: 10.1038/s41422-024-00930-7, 10.1161/CIRCRESAHA.124.325611, and 10.1161/CIRCULATIONAHA.123.065438) were identified during my initial search but were excluded as they are letters or correspondences, which did not meet our inclusion criteria focused on original research.

Re: This is exactly the problem with your exclusion criteria. These excluded research letter papers are still original research and they are peer-reviewed following the same standard as research articles. Their shorter formats simply mean they are focused on single strong points that do not need longer formats. You should still include these papers in your review.

Respond:

Thank you for this important clarification. In light of your comment, I have revised inclusion criteria and incorporated the relevant research letters into the final analysis. As a result, the total number of studies included in the review has been updated to 57.

---

## [Decision Letter · Decision Letter 2]

12 May 2025

Gene editing therapy as a therapeutic approach for cardiovascular diseases in animal models: a scoping review

PONE-D-25-07256R2

Dear Dr. Vo,

We’re pleased to inform you that your manuscript has been judged scientifically suitable for publication and will be formally accepted for publication once it meets all outstanding technical requirements.

Kind regards,

Chen Ling, Ph.D.

Academic Editor

PLOS ONE

Reviewers' comments:

Reviewer's Responses to Questions

**Comments to the Author**

Reviewer #2: All comments have been addressed

2. Is the manuscript technically sound, and do the data support the conclusions?

Reviewer #2: Yes

3. Has the statistical analysis been performed appropriately and rigorously?

Reviewer #2: Yes

4. Have the authors made all data underlying the findings in their manuscript fully available?

Reviewer #2: Yes

5. Is the manuscript presented in an intelligible fashion and written in standard English?

Reviewer #2: Yes

Reviewer #2: All my concerns have been properly addressed. I have no further comments on the content of this manuscript.

**Do you want your identity to be public for this peer review?** For information about this choice, including consent withdrawal, please see our Privacy Policy

Reviewer #2: No

---

## [Editor Report · Acceptance letter]

PONE-D-25-07256R2

PLOS ONE

Dear Dr. Vo,

I'm pleased to inform you that your manuscript has been deemed suitable for publication in PLOS ONE. Congratulations! Your manuscript is now being handed over to our production team.

Kind regards,

on behalf of

Dr. Chen Ling

Academic Editor

PLOS ONE